# A Two-Stage Framework for Automated Malignant Pulmonary Nodule Detection in CT Scans

**DOI:** 10.3390/diagnostics10030131

**Published:** 2020-02-28

**Authors:** Shimaa EL-Bana, Ahmad Al-Kabbany, Maha Sharkas

**Affiliations:** 1Alexandria Higher Institute of Engineering and Technology, Alexandria 21311, Egypt; 2Intelligent Systems Lab, Arab Academy for Science, Technology, and Maritime Transport, Alexandria 21937, Egypt; 3Department of Electronics and Communications Engineering, Arab Academy for Science, Technology, and Maritime Transport, Alexandria 21937, Egypt; 4Department of Research and Development, VRapeutic, Cairo 11728, Egypt

**Keywords:** CAD system, lung cancer, pulmonary nodules, DeepLab-V3, CNN, transfer learning, tensorflow

## Abstract

This research is concerned with malignant pulmonary nodule detection (PND) in low-dose CT scans. Due to its crucial role in the early diagnosis of lung cancer, PND has considerable potential in improving the survival rate of patients. We propose a two-stage framework that exploits the ever-growing advances in deep neural network models, and that is comprised of a semantic segmentation stage followed by localization and classification. We employ the recently published DeepLab model for semantic segmentation, and we show that it significantly improves the accuracy of nodule detection compared to the classical U-Net model and its most recent variants. Using the widely adopted Lung Nodule Analysis dataset (LUNA16), we evaluate the performance of the semantic segmentation stage by adopting two network backbones, namely, MobileNet-V2 and Xception. We present the impact of various model training parameters and the computational time on the detection accuracy, featuring a 79.1% mean intersection-over-union (mIoU) and an 88.34% dice coefficient. This represents a mIoU increase of 60% and a dice coefficient increase of 30% compared to U-Net. The second stage involves feeding the output of the DeepLab-based semantic segmentation to a localization-then-classification stage. The second stage is realized using Faster RCNN and SSD, with an Inception-V2 as a backbone. On LUNA16, the two-stage framework attained a sensitivity of 96.4%, outperforming other recent models in the literature, including deep models. Finally, we show that adopting a transfer learning approach, particularly, the DeepLab model weights of the first stage of the framework, to infer binary (malignant-benign) labels on the Kaggle dataset for pulmonary nodules achieves a classification accuracy of 95.66%, which represents approximately 4% improvement over the recent literature.

## 1. Introduction

Lung cancer remains the most prevalent cancer in the world. Because early detection can reduce the mortality rate, high-risk individuals are being screened with low-dose CT scans. Accordingly, automatic cancer recognition in CT scans has been a scope of high-paced research as it increases the efficiency of diagnosis, thus saves time, cost, and lives.

Predicting lung cancer from a chest scan has relied on reducing the high-dimensional CT scan to a few regions of interest. An example of these regions of interest is solitary pulmonary nodules (SPN) [1], which are usually an early indication of lung cancer. However, due to a large amount of information in the entire scan, relying on visual inspection can result in misdiagnosis due to the small size of the nodules and the diverse sources of human error. Although lung cancer late diagnosis rates represent 70–80% of the patients, we can increase early diagnosis by improving the quality of pulmonary nodule detection (PND), which enhances the chance of successful surgical treatment. This has become attainable with the evolution of artificial intelligence and the ever-growing potential of deep neural network (DNN) models in computer vision and medical image analysis, which is a cornerstone for Computer-aided diagnosis (CAD) systems [2].

CAD systems have evolved from the X-ray film, which was used for early research on lung cancer. X-ray imaging depends on each detection site’s density; therefore, it often misses small nodules and the nodules hidden behind other body organs and blood vessels, resulting in reduced detection results. With the continuous development of imaging technologies, low-dose CT scanning has been gradually showing high potential. It can detect even millimeter-sized tumors, and has become one of the most reliable techniques for identifying lung nodules in the early stages of lung cancer.

There has been an ever-growing body of research on automated detection of pulmonary nodules in medical images. Arguably, the literature can be divided into two families of algorithms: feature engineering-based methods and deep learning-based methods [3]. The literature of the former family of modules features a wide variety of lung segmentation and nodule detection techniques. The impact of different image preprocessing and postprocessing (false-positive reduction) methods was also extensively analyzed and discussed in the literature. On the other hand, deep learning-based methods have involved different N-D neural network structures, learning procedures, and data augmentation methods, in addition to others [4]. One of the promising deep network models in the recent literature, that has been adopted in various medical applications is DeepLab [5,6].

In this research, we propose a two-stage framework for automated PND in CT scans. Particularly, we show that feeding the output of an effective semantic segmentation method to a localization-classification stage would achieve an accuracy that is comparable, and can outperform the state of the art techniques. In the first stage, we adopt a deep learning approach that employs the DeepLab-V3 [7] model, which is composed of an encoder–decoder with Atrous Separable Convolution, for the semantic segmentation of the candidate’s nodules [8]. This is combined with transfer learning [7] to save the training time and improve efficiency. In stage two, a procedure is developed to localize the nodules in the input images, then classify whether the localized region is a nodule or not. We compared the proposed pipeline with the recent literature objectively, and we present very promising results with regards to accuracy and sensitivity on the widely-adopted LUNA16 dataset. We also used the weights of the models trained on LUNA16 to infer binary labels (malignant-benign) labels on the Kaggle dataset for pulmonary nodules. It starts by detecting the nodules in the input images, then classifying the images to either malignant or benign. We show that this transfer learning approach has attained a performance that is comparable to, and provides an improvement on the recent literature. The key contributions of this paper can be summarized as follows.

We propose to employ the recently published DeepLab neural network model in the semantic segmentation of pulmonary nodules in CT scans. We show that it significantly outperforms other existing models in terms of the attained Jaccard index and the Dice Coefficient.We analyze the performance of the DeepLab model under two network backbones, namely, Xception [9] and MobileNet-V2 [10], which suit server-side and client-side applications, respectively. We completed performance analysis by presenting the impact of diverse model parameters on the quality of the output segmentation. We also study the influence of preprocessing the CT scans (notably, the initial lung segmentation) on the nodule segmentation.We propose a two-stage approach for automated pulmonary nodule detection in CT scans, which is comprised of a nodule localization and classification stage after the semantic segmentation stage. We show that the performance of the proposed system is of comparable performance to state of the art, and outperforms several existing techniques. We also show that every stage in the proposed approach contributes to the collective accuracy of the system.Despite missing nodule-level labels from the kaggle dataset, we introduce a new approach for generating such labels using transfer learning, based on the model weights that we learned from the LUNA16 dataset. To the best of our knowledge, this is a new approach for *synthesizing* missing labels for that dataset, which further empowers the researchers to compare their proposed frameworks on datasets of diverse characteristics.

The rest of the paper is organized as follows. Related work that used deep learning methods on different lung datasets is presented in Section 2. In Section 3, the proposed two-stage framework for pulmonary nodule detection and malignancy classification is discussed. Experimental results are given in Section 4, and finally, the work is concluded in Section 5.

## 2. Related Work

The tremendous evolution of computer hardware, especially graphics processing units (GPUs) and multi-core processors, has stimulated high-paced advances in deep learning. Deep learning (DL) is a subset of machine learning (ML), and it involves the construction and training of deep neural networks (DNNs) that are capable of learning in a supervised or unsupervised manner. The success of deep learning algorithms in diverse computer vision applications has strongly inspired its adoption in medical images analysis. The increasing demand of medical records digitization affords many types of scans/datasets to study.

One of the widely-adopted NNs in medical image analysis is the U-Net model, which is a convolutional neural network that was proposed by Olaf Ronneberger et al. [11]. Initially, it was proposed for medical image segmentation using an encoder–decoder structure in electron microscopic stacks, and it outperformed other methods in the 2015 International Symposium on Biomedical Imaging (ISBI) competition. Tian Lan et al. [12] proposed a RUN network to complete the detection of the nodule in a single step by introducing a residual network shortcut to improve the traditional U-Net. They validated their method on the LUNA16 dataset, and they achieved a sensitivity of 90.90% at two false positives per scan and 71.9% of the dice coefficient. Mostafa Salem et al. [13] used U-Net to synthesize lesions in MRI images. His evaluation was based on (1) measuring the similarities between the real and the synthetic images and (2) lesion detection performance by segmenting both the original and synthetic images using a state-of-the-art segmentation framework.

Kaiming He et al. [14] proposed the Residual network framework to ease the training of networks. They obtained a 28% relative improvement on the COCO object detection dataset, and they won the ImageNet Large Scale Visual Recognition Competition (ILSVR). Jin et al. [15] used the 3D-AlexNet to classify the images of the DSB Kaggle lung CT scan. They compared the 3D-AlexNet architecture with various input sizes and a different number of epochs. Experimental results showed an accuracy rate of 87.5%. Huseyin et al. [16] proposed two Convolutional Neural Network (CNN)-based models to diagnose lung cancer on CT images. They are named Straight 3D-CNN, and they adopted conventional softmax and hybrid 3D-CNN with Radial Basis Function (RBF)-based SVM. The hybrid 3D-CNN with SVM achieved results (91.81%, 88.53%, and 91.91% for accuracy rate, sensitivity, and precision, respectively). Nasrullah, N. et al. [17] proposed two deep 3D networks called (CMixNet) Architectures for lung nodule detection and classification. They used faster RCNN learned on features from CMixNet and U-Net for Nodule detections. Classification of the nodules was performed through a gradient boosting machine (GBM) on the learned features from the designed 3D CMixNet structure. They evaluated their proposed system on LIDC-IDRI datasets achieving a sensitivity of (94%) and specificity of(91%). Hwejin Jung et al. [18] introduced a method for nodule classification, which uses three dimensional deep convolutional neural networks (3DCNNs) and checkpoint ensemble method to discriminate nodules between non-nodules, achieving a CPM score of 0.910.

Liang-Chieh Chen et al. [19] took advantage of deep CNNs performance in image classification and object detection, combining it with probabilistic graphical models for addressing the problem of semantic image segmentation. They named their system Deeplab, reaching 71.6% IOU accuracy at the PASCAL VOC-2012 semantic image segmentation task. Followed by the authors of [20], which highlights convolution with Atrous convolution, allowing control of the resolution of feature responses computation within CNN explicitly. Their improvement allowed them to reach 79.7% mIOU on the same dataset. Afterwards, they proposed improvement [21] over atrous convolution by adopting multiple atrous rates to handle the problem of segmenting objects at multiple scales called Deeplab-v3. Finally, they proposed DeepLab-V3 plus [7], which extends the previous version with a capable yet simple decoder module; therefore, the experimental results improved to reach 89% without any postprocessing.

Deeplab ability to reach outstanding results makes it rapidly spread in many applications like with Kaijian Xia et al. [22] who proposed an improved model based on the basic framework of DeepLab-V3, and Pix2pix network is introduced as the generation adversarial model for Liver Semantic Segmentation. Manu Goyal et al. [23] introduced a method for ROI detection for a skin lesion using FRCNN-Inception-V2 and SSD-Inception-V2 and they compared the performance of skin localization methods with the state-of-the-art Deeplab segmentation method. Also, Masood, A. et al. [24] proposed a computer-assisted decision support system based on the DFCNet model for classification of each detected pulmonary nodule into four lung cancer stages.

## 3. Methods

This paper presents a two-stage system for nodule segmentation, localization, and classification. Figure 1 depicts the stages of the proposed system. In the first stage, we employ the DeepLab-V3 plus model for nodules semantic segmentation. Towards this goal, we examine the performance of the following approaches. (1) By means of preprocessing, we segment the ROI, i.e., the lungs in our case, in the slices before feeding them to the DeepLab model. This means that we narrow down the ROI to the lungs before performing semantic segmentation to detect the nodules. (2) We feed the images to the DeepLab model without preprocessing, i.e., blind semantic segmentation. Note that in stage 1, only LUNA16 is used for training.

The flow of the second stage, though, differs depending on the adopted dataset. In stage 2-a (in Figure 1), the segmented slices of LUNA16 are fed to the localization-classification model, which produces binary (nodule/non-nodule) labels accordingly. However, in stage 2-b, the slices of the Kaggle dataset are semantically segmented first, using the semantic segmentation weights learned from LUNA16 (transfer learning), before they get passed to the localization-classification model. In stage 2-a, we applied two object localization and classification architectures, namely, FRCNN and SSD, with Inception-V2 as the backbone. In stage 2-b, though, we compared two classification architectures, namely, inception-V3 and mobileNet-V1. For both stages, 2-a and 2-b, we use data augmentation techniques to generate a larger number of training examples to avoid overfitting.

### 3.1. Datasets

In our experiments, we used the following datasets.
Lung Nodule Analysis 2016 challenge (LUNA16) dataset https://luna16.grand-challenge.org/: The LUNA16 dataset is acquired from the Lung Image Database Consortium and Image Database Resource Initiative (LIDC-IDRI) database [25]. This challenge contains 888 CT scans. The data includes the CT scans and nodules labels (list of nodule center coordinates and diameter) for each patient. Each radiologist marks the lesions in the lung as non-nodule if the nodule is smaller than 3 mm and as nodules if greater than or equal to 3 mm. As part of this dataset, we are only considering nodules that are >= 3 mm and that are accepted by three out of four radiologists. The dataset contains nodule annotations without the cancer status of patients. LUNA16 data is used to train the Deeplab-V3 plus model for nodule detection.Lung CT scan dataset from Kaggle’s Data Science Bowl (DSB) 2017 https://www.kaggle.com/c/data-science-bowl-2017: The dataset contains 1397 labeled patient data. Each patient has CT scan slices and a label (1 for a malignant case and 0 for benign one). The CT scan data consists of a various number of axial slices with a size of 512 * 512 (typically from 100 to 400 slice). Figure 2 shows sample slices for a single patient. Note that the Kaggle dataset does not have the locations or sizes of labeled pulmonary nodules within the lung, so we are using LUNA dataset segmentation results to obtain this missing detection information.

We utilize both datasets to train our system. A sample slice difference from one patient for both datasets is shown in Figure 3.

### 3.2. Preprocessing and Lung Segmentation

We perform lung region segmentation as a preliminary phase due to the heterogeneity in the lung region, and due to the homogeneous densities in pulmonary structures such as arteries, veins, and bronchi as well. Under CT scanning, the lung area primarily includes air, which has a value of approximately −1000, and that for the lung is approximately −500, in the Hounsfield unit. Table 1 demonstrates the Hounsfield scale, which supports radiologists to analyze CT scans [15,26].

At this stage, for the LUNA16 dataset, the image is segmented by removing the background. The lung segmentation is a challenging issue because of the heterogeneity in lung region and homogeneous densities in pulmonary structures such as arteries, veins, and bronchi. The thresholding method can be applied to the lung-specific level, followed by morphological operation methods, including 3D connected component analysis, region erosion, region dilation, and lung volume thresholding for lung segmentation. Then, an improved image of the lung will be produced [27,28]. After these preprocessing steps, a patient’s lung volume and its associated label are in pair, ready for training (https://github.com/booz-allen-hamilton/DSB3Tutorial).

### 3.3. Stage 1: Deeplab-V3 Plus for Lung Nodule Detection and Semantic Segmentation

After image preprocessing, we detect the nodules on this reduced area. The task of detecting nodules is very challenging because the nodules can be placed either inside the lungs or on the walls, and they are are very hard to discriminate from shadows, vessels, and ribs. First, we extract the nodules regions from the LUNA dataset in images as a mask, as shown in Figure 4. The masks and segmented lung images are fed into a CNN. The input to the model is a single slice, and the output is the corresponding ground truth binary mask showing nodule locations. We are using Deeplab-V3 plus that is described in Figure 5. Google has extended DeepLab-V3 plus to include a simple decoder module to enhance the results of segmentation, mainly along the boundaries of the object.

In this paper, we employ that extended model in a medical application, which is the lung nodule segmentation. We briefly explain the stages of this model as follows.

**Atrous convolution:** Operates with space inside the filter that can take a large field of view, it is adaptively modified by varying the rate value [29,30] as in Figure 6. With a case of two-dimensional signals, atrous convolution is applied over the input feature map *x* as described in Equation (Equation 1) for each location *i* on the feature map output *y* and a convolution filter *w*:
(1)$$y\left[i\right]=\sum_{k}\left[x_{i}+r.k\right]w\left[k\right]$$
where the atrous rate *r* determines the stride with the sampled input signal, standard convolution is a particular case with r=1.**Depthwise separable convolution:** This process consists of two operations, a depthwise convolution followed by point-wise convolution (i.e., 1 * 1 convolution). It drastically reduces the computational complexity. The depthwise convolution independently realizes a spatial convolution for each input channel, then the output from the depthwise convolution is combined with point-specific convolution [9].**Encoder:** This stage is important to extract the essential information from the images by implementing a pre-trained model. The essential information for segmentation tasks is the objects in the image and their locations. We used the following two models as the primary feature extractor.
(a)**Modified Xception-65:** Xception [9] is an expansion of the Inception architecture replacing conventional Inception modules with separable depth convolutions. We train DeepLab-V3 plus using Aligned Xception as its main feature extractor, where all the maximum pooling operations are replaced by depthwise separable convolution with striding. After each 3 × 3 depthwise convolution, extra batch normalization and ReLU activation are joined, and the depth of the model is increased without changing the entry flow network structure [31,32].(b)**Mobilenet-V2:** In MobileNet-V2 [10], the inverted residual structure is implemented in narrow layers without nonlinearities. It is used as a backbone to extract features with performance results that are accomplished for semantic segmentation and object detection. This model is built on the concepts of MobileNet-V1 [33], using depthwise separable convolution as efficient building blocks. It has two new features introduced to the network: (1) linear bottlenecks between the layers and (2) shortcut connections between the bottlenecks.**Decoder:** The extracted information from the encoding phase is used to generate an output with the original input image size [7], as shown in Figure 5.

### 3.4. Stage 2-a: ROI (Lung Nodule) Localization and Classification of LUNA16 Dataset

For our ROI detection, we use two models: Faster RCNN-Inception-V2 model [34] and Single Shot Multibox Detector(SSD)-Inception-V2 model [23,35]. In general, our method consists of three stages to produce ROIs using FRCNN-Inception-V2:**Feature extraction:** The input is image fed to CNN to extract feature maps on the last convolutional layer. We used Inception-V2 [36], which is built to scale up CNN to maintain a high-performance visionary system with a modest computational cost.**Proposals generation:** We ran the sliding window on the feature maps from the previous stage, resulting in several proposals with different aspect ratios and the same center.**Classification and bounding box regression:** The proposals from the previous stage are classified by the mean of CNN and refined further to get the final ROI [37]. These stages are depicted in Figure 7.

The SSD Single Shot Multibox Detector [35] is a recent object localization meta-architecture based on a bounding box regression principle. In comparison with the Faster R-CNN, it utilizes a single-stage feedforward for predicting anchor boxes and class scores for the objects in each box. Therefore, the anchors generated are much faster than other object localization models, making it more acceptable to mobile platforms with reduced hardware settings than computers. Finally, each region’s classification scores are calculated based on the score obtained in the previous step. For this model, we propose using Inception-V2 [36] as a base network to extract features and classify anchor boxes with all meta-architectures for the detection of ROI nodule as shown in Figure 8.

### 3.5. Stage 2-b: Malignancy Classifiers of Kaggle Dataset

Once we trained the Deeplap-V3 plus [7] on the LUNA16 data, we run it on 2D slices of Kaggle dataset for nodule detection. However, as shown in Figure 9, Deeplab-V3 plus produces a strong signal for the actual nodule but also produces a lot of false positives, which would cause problems when training the classifier in the following stage of the pipeline.

To avoid this, we use a method that involves keeping nodule detections that were found in regions where multiple detections were made across all slices for a single patient. We generate a heatmap to combine the detections of the Deeplab-V3 plus from on all the slices for a single patient, i.e., high-confidence detections. Figure 10 shows all the nodule regions detected for a patient, such that the more reddish/bright the regions, the more detections at those pixel locations. We can see that in various locations of the CT scan, there are false positive detections. However, only a few regions have multiple detections, i.e., red blobs with higher intensity. The heatmap is then thresholded to keep only the pixels that have repeated detections. The idea behind this approach is that the regions that have multiple detections from the nodule segmentation are more likely to be actual nodules. From there, only the slices that overlap with the thresholded heatmap nodule regions are kept. The data obtained from these nodule detections will then be used to train a classifier to predict whether or not a person has cancer, i.e., malignant-benign labeling (https://github.com/olinguyen/kaggle-lung-cancer-detection). We use two convolution neural networks (CNN): the MobileNet-V1 [33] and Inception-V3 [36] models which are designed for image recognition tasks [38].

Inception-V3 [36] network model is a deep neural network, and it is challenging for us to train it directly with a low configuration computer, as it takes at least a few days to train. Tensorflow [39] provides a way to use the transfer learning to retrain the final layer of the inception for new categories. We use the transfer learning method that decreases the time taken to train from scratch by taking a fully trained model for a set of categories such as ImageNet and retraining for new classes from existing weights. In our experiment, we retrained the final layer from scratch, while leaving all the rest untouched. The number of output nodes in the last layer is equal to the number of classes in the data set. For example, the ImageNet dataset [38] has 1000 classes, so in the original Inception-v3 model, the last layer has 1000 output nodes. Inception-v3 is comprised of two stages: feature extraction stage with a convolutional neural network (CNN) and Classification stage with fully-connected and softmax layers [40].

MobileNet-V1 [33] use depth-wise separable convolutions [9]. The main difference between the MobileNet architecture and a standard CNN’s is that instead of a single 3 × 3 convolution layer followed by the batch norm and ReLU, MobileNets split the convolution into a 3 × 3 depthwise conv and a 1 × 1 pointwise conv. MobileNets have two parameters that we can tune: width multiplier and resolution multiplier. The width multiplier helps us to thin the network, whereas the resolution multiplier adjusts the input dimensions of the image, decreasing the internal representation at each layer.

## 4. Results and Discussion

Our experiments were implemented on a machine with a GeForce GTX 1080 Ti GPU (8 GB VRAM). Tensorflow was used as the backbone in all experiments, and Python was the primary programming language. With regards to the performance metrics, the most commonly used metrics in the literature of segmentation and classification of medical images are show in Table 2, where TP, FP, FN, and TN refer to the number of True Positives, False Positives, False negatives, and True Negatives, respectively. Our quantitative analysis also employs the *mAP@.50IoU* and the *mAP@.75IoU*, which refer to the Mean Average Precision at 50% IoU and 75% IoU, respectively.

To avoid overfitting, we use data augmentation techniques to generate a large number of training scans. Typically, the training stage of convolutional neural networks requires a large number of positive and negative samples to increase the training data and reduce the class imbalance. Accordingly, we use the rotation and flipping both horizontally and vertically to generate more lung nodules for model training. After augmentation, we got 3258 detected nodules from the DeepLab model and 10,000 thresholded nodules from the Kaggle dataset. Results on LUNA16 and Kaggle’s datasets are presented in Section 4.1 and Section 4.2, respectively.

### 4.1. Results on LUNA16

First, the images were resized to 512 × 512, and then converted to TFRecord to make the dataset compatible with the DeepLab code. Various options regarding network models were available, including MobileNet-V2 and Xception. Also, different hyperparameters are tunable, including the learning rate, the weighting of foreground and background, and the weight decay. In our experiments, the weights of the checkpoint used to segment the Pascal dataset were used as initial weights, with 0.0001 learning rate and foreground loss weight 100 for 50 K steps. In Figure 11, we visualize the output results on the LUNA16 validation set with segmented images (preprocessing step) in Figure 11a, and without preprocessing in Figure 11b. The 4th row in the figure shows some examples of false positive nodules.

Afterwards, to deal with the false positive nodules, we trained a Faster-RCNN Inception-V2 to detect and classify the nodules. We used a batch size of 24 with 10,000 steps, a Momentum optimizer value of 0.9 with step learning rate manually set to 0.0002 of the initial rate. We also trained a SSD Inception-V2 model with a batch size of 24, 10,000 steps, a RMS optimizer with 0.95 decay factor, and a learning rate of 0.004. We used a network that is pre-trained on the MS-COCO dataset—a 90-object class dataset that consists of more than 80,000 images. In Figure 12, we report some examples of the inference generated by the Faster-RCNN Inception-V2 nodule localization model.

In Table 3, we compare the performance of each trained model (Xception-65 and MobileNet-V2) on LUNA16, combined with DeepLab-V3 plus, for the semantic segmentation stage with different variants of the U-Net model. On the validation set, we achieved a 79.1% mIOU on the preprocessed (parenchyma segmented) images, which represents 60% increase compared to U-Net. We have also achieved a 88.34% Dice Coefficient, which represents a 30% increase compared to U-Net. Note that we also experimented the performance without preprocessing the LUNA16 slices, and we attained a 78.02% mIOU—a result that may motivate computational savings by dropping the segmentation preprocessing step, as the gain is only 1% increase in the mIoU.

As mentioned in Section 3.3, Xception-65 and MobileNet-V2 [9,33] encoders are used as backbones with DeepLab-V3 plus. As shown in Table 4, significant runtime savings and, simultaneously, significant mIoU gains can be achieved using DeepLab-V3 plus (with Xception-65 and MobileNet-V2 encoders) compared to U-Net. MobileNet-V2, which is a fast network structure designed for mobile phones, takes 2.45 h per 50k steps with 72.73% mIoU, whereas the Xception model, which is a robust network framework designed for server-side deployment, takes 3.45 h for 20k steps with 78.77% mIoU. On the other hand, U-Net model takes 8.88 h for training only 20 epochs achieving only 49.25%. Additionally, in Table 5, we report different output stride with the corresponding mIoU results, where the output stride is the ratio of the size of the original image to the final encoded features size—a parameter that strongly influences the adopted encoder.

Following the semantic segmentation stage, we report the results obtained from the localization-classification stage. First, the performance comparison of our adopted model (DeepLab-V3 plus(ex_65) + FRCNN-Inception-V2) with other models is shown in Table 6. It features the highest sensitivity in nodule classification with 96.4%, and lowest false positive per scan (FP/Scan) of 0.6 compared to other models. We also report the TN and FN performance of the Faster RCNN model by the confusion matrix in Table 7. Finally, Table 8 shows a comparison between the FasterRCNN-Inception-V2 and SSD Inception-V2 models, and it indicates the superiority of the FRCNN Inception-V2 with regards to the average recall (AR), the classification loss, and the localization loss. However, SSD shows significantly faster (almost 100 times faster) inference than FRCNN.

### 4.2. Results on Kaggle’s Dataset

As mentioned in Section 3.5, we generate a heatmap to combine the detections from all slices of a single patient on Kaggle’s dataset. In this operation, we evaluated several values for thresholding the heatmap, in order to keep only the pixels that represent repeated detections, i.e., high-confidence detections, as shown in Figure 10. Table 9 shows the different threshold values with the corresponding validation accuracy results using the Inception-V3 model.

Training was carried out using nearly 10K augmented images with a batch size of 100 for 10K steps/iterations using the Inception-V3 and MobileNet-V1 models. For transfer learning on the Inception-V3 model, we removed the weights of last layer and re-trained it using Kaggle’s dataset. The number of output nodes that is equal to the number of the classes is modified to be 2. The last layer of the model is trained with a backpropagation algorithm and the weight parameter is adjusted using the cross-entropy cost function by calculating the error between the softmax layer output and the specified sample class label vector. For the MobileNet-V1 model, we fine-tune the following parameters. (1) Width Multiplier with 1.0, 0.75, 0.50, and 0.25. (2) Resolution Multiplier with 224, and we found that accuracy drops off smoothly from width multiplier = 1 to 0.5 until the value of 0.25. During the run, the following outputs are generated showing the progress of algorithm training.

**Training accuracy:** Is the percentage of the images that were labeled correctly in the current training batch.**Validation accuracy:** Is the accuracy (percentage of correctly labeled images) of a randomly selected group of images from a different set.**Cross entropy:** Is defined as a loss function that informs us how well the process of learning progresses, formulated as follows:
(2)$$Cross\_entropy\_loss=-\sum_{c=1}^{M}y_{o,c}log\left(P_{o,c}\right)$$
where *M* is the number of classes (M=2 in our experiment for two classes: Malignant, Benign), *y* is a binary indicator if class label *c* is the correct classification for observation *o*, and *P* is a predicted output probability observation *o* of class *c*.

For the classification of the detected nodules from the Kaggle dataset, the results that are shown in Table 10 compares the Inception-V3 model with the MobileNet-V1 model. After 10,000 steps, we achieved the highest accuracy value of 96.05% and 95.66% for the training and validation accuracy, respectively, using the Inception-V3 model. The Inception-V3 model also achieved the lowest cross-entropy loss of 0.166 for the validation set. We show the variations of the accuracy and cross-entropy for that model in Figure 13. For MobileNet, the width multiplier = 1 and the resolution multiplier = 224 achieved an accuracy of 94% and 93% for training and validation, respectively, outperforming other parameter settings for MobileNet.

Furthermore, we also compared the performance of the adopted architectures (Inception-V3 and Mobilenet-V1) with a linear classifier, vanilla 3DCNN, 3D Googlenet, 3D-AlexNet, DFCNet [24], TumorNet [42], CMixNet [17], and straight and hybrid 3D CNN architectures [16]. Table 11 presents a summary of the accuracy, sensitivity, and specificity of all the architectures.

Considering the results in Table 11, we demonstrate that the adopted architecture (Inception-V3), as a part of the proposed two-stage framework, surpassed all other architectures by achieving a 95.66% accuracy. Note that although 3D CMixNet + GBM + Biomarkers model achieved higher accuracy than our Mobilenet-based model, it used additional features which are of physiological symptoms and clinical biomarkers. Moreover, compared to other methods, the proposed framework with the two adopted architectures (Inception and MobileNet) achieved the highest and the second-highest specificity. With regards to sensitivity, though, CMixNeT outperformed the proposed framework; nevertheless, this was attained on a different dataset. The results presented in this section prove that Inception-V3 and MobileNet classifiers, when combined with semantic segmentation and transfer learning, can improve significantly the performance of deep learning models in classifying 3D lung CT scan images.

## 5. Conclusions

In this paper, we proposed a two-stage framework that employs deep learning frameworks for PND in CT scans, trained and tested on LUNA16 and Kaggle 2017 datasets. We showed that adding a semantic segmentation stage before localizing and classifying the nodules can remarkably improve the accuracy. We proposed the recently proposed DeepLab-V3 plus model for semantic segmentation, and we demonstrated that it is capable of attaining state-of-the-art Dice Coefficient and IoU. On the LUNA 16 test set, we achieved a 30% increase and 60% increase in the Dice Coefficient and the mIoU, respectively, compared the standard U-Net and its more recent variants. We adopted two encoders as feature extractors, Xception-65 and MobileNet-V2, and we demonstrated significant runtime reductions and mIoU gains with both encoders compared to U-Net. The second stage of the framework is meant to handle the false positives resulted from the first stage. We fed the output of the semantic segmentation to FRCNN Inception-V2 and SSD Inception-V2 models for localizing and classifying the nodule lesions, and we showed that the former’s sensitivity is comparable to the state-of-the art. We have also extended the usage of the deep models trained on LUNA16 to infer binary (malignant-benign) labels on Kaggle’s dataset. Particularly, although missing nodule-level labels in the original dataset, we used the LUNA16-trained DeepLab-V3 plus to generate initial nodule detections. This was latter followed by a localization-classification stage, similar to the procedure carried out on LUNA16. The proposed framework attained an eventual accuracy and specificity of 95.66% and 97.24%, respectively, on Kaggle dataset, and a FP/Scan of 0.6, outperforming other recent models in the literature, including deep models. Future directions include the adoption of knowledge distillation, in the pursuit of the combination of effectiveness and speed. We also look forward to evaluating the effectiveness of the learned models across different imaging modalities and in other medical applications.

## Figures and Tables

**Figure 1 diagnostics-10-00131-f001:**
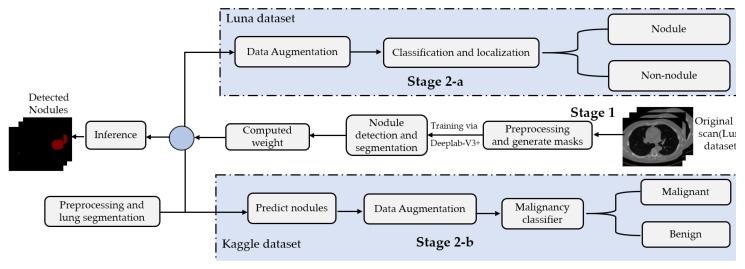
The block diagram of the proposed method. Stage 1 involves employing DeepLab-V3 plus for semantic segmentation. Stage 2-a involves the localization and the classification of nodules on LUNA16. Stage 2-b involves the detection and the classification of nodules on Kaggle’s dataset.

**Figure 2 diagnostics-10-00131-f002:**
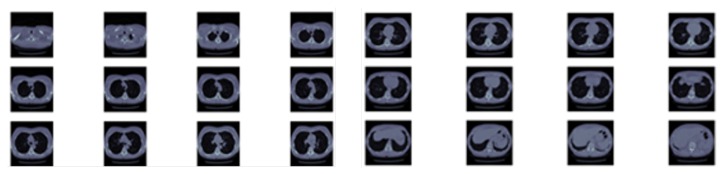
A sample slices for a single patient of the Kaggle dataset.

**Figure 3 diagnostics-10-00131-f003:**
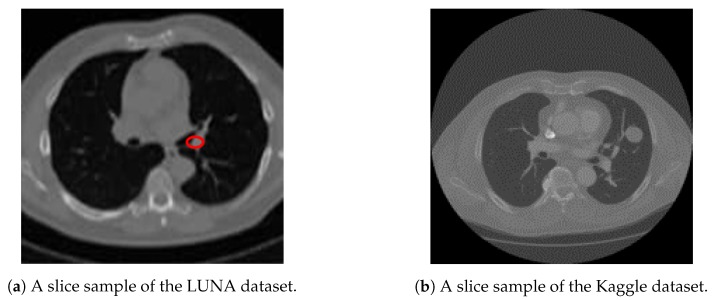
A sample slice from both Kaggle and LUNA datasets, showing a cross section of a patient chest cavity. Figure 3a shows a LUNA slice with an annotated nodule. Figure 3b shows that no annotations are given about nodule positions or sizes for Kaggle dataset.

**Figure 4 diagnostics-10-00131-f004:**
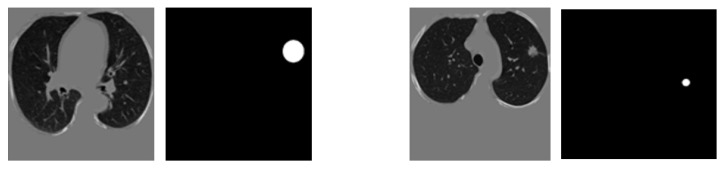
Two examples for a training pair: the input images after segmenting the parenchyma and the ground-truth binary nodule mask.

**Figure 5 diagnostics-10-00131-f005:**
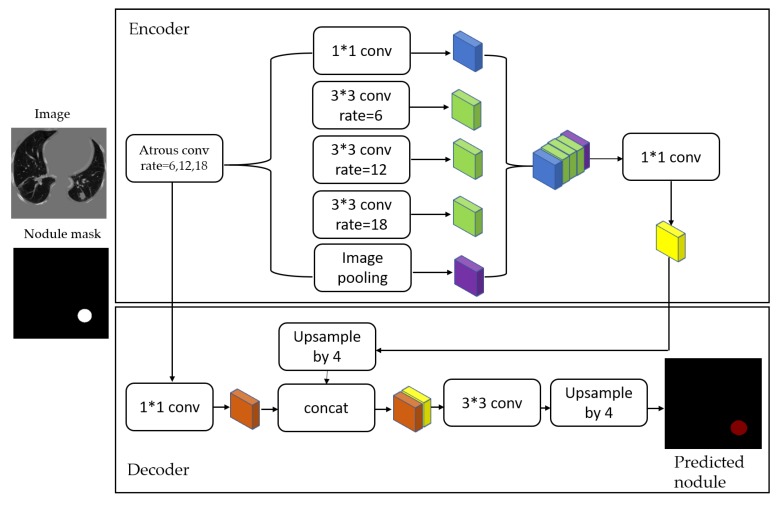
Pulmonary nodule detection and segmentation framework. First, segmenting the lung parenchyma from the raw lung CT images and then using preprocessed images to train the Deeplab-V3 plus network by employing an encoder–decoder structure. This figure was inspired by work in [7].

**Figure 6 diagnostics-10-00131-f006:**
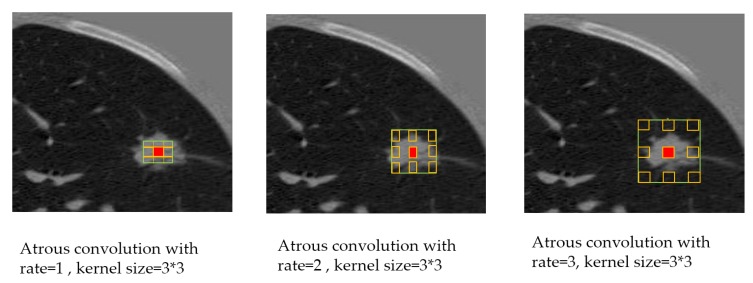
Explanation of the Atrous Convolution with different rates indicating an enlarged field of view.

**Figure 7 diagnostics-10-00131-f007:**
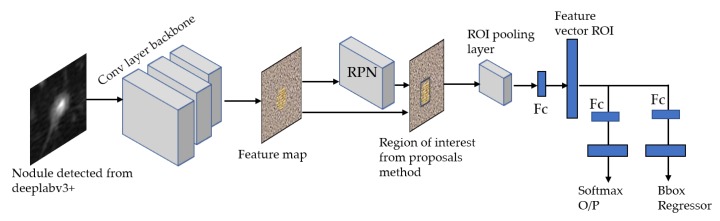
Proposed FastrRCNN model for region-of-interest (ROI) lung nodule localization and classification using Inception-V2 as a backbone.

**Figure 8 diagnostics-10-00131-f008:**
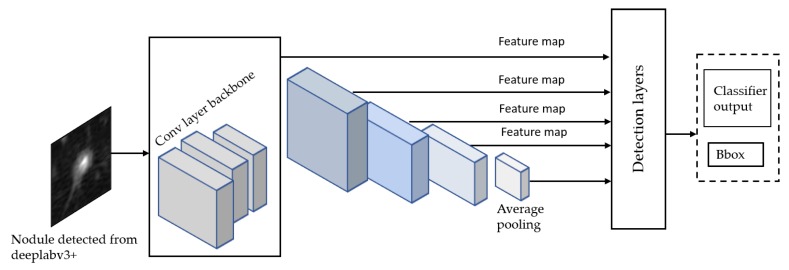
Proposed SSD model for ROI lung nodule localization and classification using IncepttionV2 as a backbone.

**Figure 9 diagnostics-10-00131-f009:**
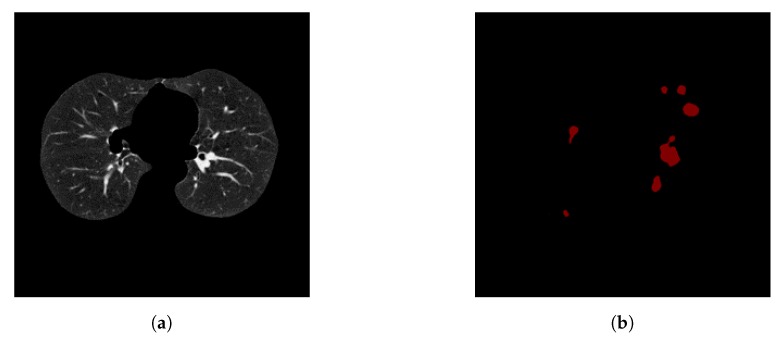
Output nodule detection: Figure 9a shows the sample input from kaggle test set to the Deeplab model, and Figure 9b shows the Deeplab predicted output with a lot of false positives nodules.

**Figure 10 diagnostics-10-00131-f010:**

Left: raw CT scan. Middle: a patient’s heatmap generated from all the slices. Right: thresholded heat map to keep regions with multiple detections.

**Figure 11 diagnostics-10-00131-f011:**
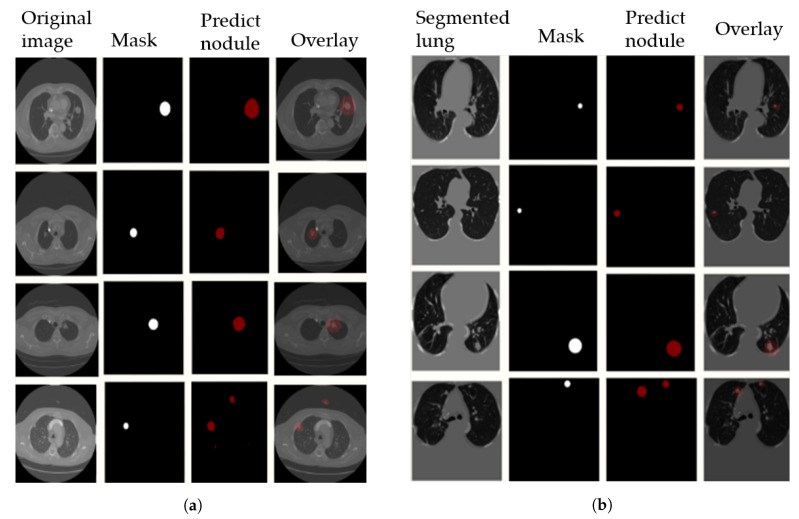
Output results on Luna16 Val set with segmented images. Figure 11a original images without preprocessing step, and Figure 11b preprocessed images(segmented lung input). A sample of the segmentation prediction results and its overlay (right of image) are compared to ground truth masks.

**Figure 12 diagnostics-10-00131-f012:**
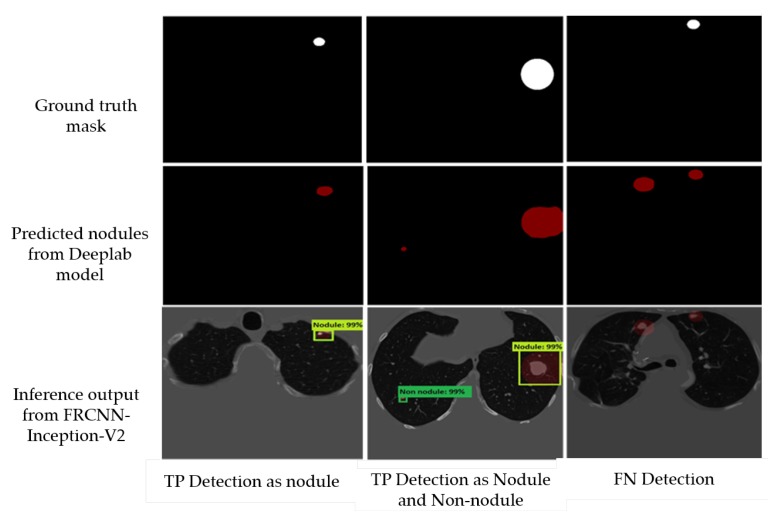
Examples of the inference produced by the Faster-RCNN Inception-V2. The first column shows TP detection as a nodule with IOU = 99%. The second column shows TP detection as a nodule with IOU = 99% and a FP with IOU = 99%. Finally, the third column shows a FN where the model failed to detect the nodule lesion.

**Figure 13 diagnostics-10-00131-f013:**
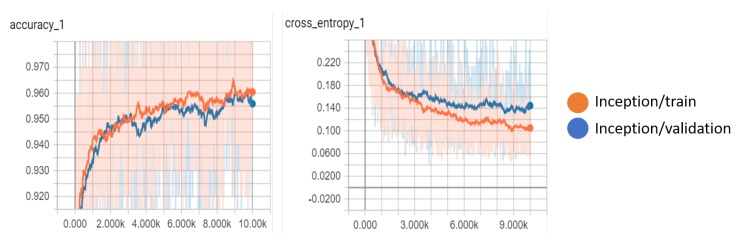
The variation of accuracy and cross-entropy on the kaggle dataset using the Inception-V3 model.

**Table 1 diagnostics-10-00131-t001:** Hounsfield scale.

Substance	Air	Lung	Fat	Water	Blood	Soft Tissue	Bone
HU	−1000	−500	−100 to −50	0	+30 to +45	+100 to +300	+700 to +3000

**Table 2 diagnostics-10-00131-t002:** Quantitative performance measures.

Metric	Equation
Intersection over Union (IoU)	TP/(TP + FP + FN)
Dice coefficient (DCS)	2TP/(2TP + FP + FN)
Accuracy	(TP + TN)/(TP + FP + TN + FN)
Specificity	TN/(TN + FP)
Precision	TP/(TP + FP)
Recall = Senstivity	TP/(TP + FN)

**Table 3 diagnostics-10-00131-t003:** Comparing the attained Dice Coefficient and the mIoU by different variants of U-Net vs. the proposed methods.

Network Structure	Dice Coefficient	mIoU
Dual-Path Residual U-Net	63.05%	46.03%
Traditional U-Net	66%	49.25%
RUN: Residual U-Net	71.9%	56.13%
Ours: DeepLab-V3+ with X-65 encoder	88.34%	79.1%
Ours: DeepLab-V3+ with MobileNet-V2 encoder	84.2%	72.7%

**Table 4 diagnostics-10-00131-t004:** Comparison of runtime and mIOU attained by U-Net and the models adopted in this research.

Backbone	mIOU	Runtime	Number of Steps/Epochs
Xception-65	78.77%	3.45 h	20k steps
Mobilenet-V2	72.73%	2.45 h	50k steps
U-Net	49.25%	8.88 h	20 epochs

**Table 5 diagnostics-10-00131-t005:** Different output stride at Val set with Xception-65.

Output Stride	8	16	32
mIOU	76.9%	79.1%	67.9%

**Table 6 diagnostics-10-00131-t006:** Performance comparison for DeepLab-V3 plus(ex_65) + FRCNN Inception-V2 and other models on the LUNA16 dataset.

Model	Senstivity	FPs/Scan	Accuracy	Specificity
CNN [24]	70.23%	4.7	74.01%	79.47%
DFCNet [24]	73.14%	4.2	80.12%	81.95
RUN:Resdiual-U-Net [12]	90.9%	2	–	–
D48 [18]	91.3%	1.6	94.8%	98.4%
3D CMixNet Faster-RCNN [17]	90%	1.1	–	–
DeepNet [41]	84.8%	1	–	–
ResNet [41]	86.7%	1	–	–
ResNet+HL [41]	90.5%	1	–	–
ESB-ALL [18]	93.3%	0.7	96.3%	99.3%
Ours Deeplab-V3 plus(ex_65) + FRCNN-Inception-V2	96.4%	0.6	97%	99.4%

**Table 7 diagnostics-10-00131-t007:** The confusion matrix of the Deeplab-V3 plus(ex_65) + FRCNN-Inception-V2 model.

		Predicted	
		*Nodule*	*Non-Nodule*
**Actual**	*Nodule*	0.964	0.036
	*Non-Nodule*	0.006	0.994

**Table 8 diagnostics-10-00131-t008:** Performance comparison between Faster RCNN and SSD models using InceptionV2 as a feature extractor.

Model	FRCNN	SSD
Backbone	Inception-V2	Inception-V2
# scans	3258	3258
MAP(0.5)	1.0000	0.812202
MAP(0.75)	0.93995	–
AP(0.5)	1.0000	Nodule = 0.914, Non Nodule = 0.7096
AP(0.75)	0.940	–
AR(0.5:0.95)	0.964	0.72
Classification loss	0.011148	1.582611
Localization loss	0.009361	0.602215
Test image(sec/image)	5.05	0.05

**Table 9 diagnostics-10-00131-t009:** Different heatmap threshold values with validation accuracy results using the Inception-V3 model.

Heatmap Threshold	180	190	200	210
Validation accuracy	93.9%	95.66%	95.1%	91.5%

**Table 10 diagnostics-10-00131-t010:** Comparison of the Inception-V3 model with Mobilenet-V1 model with regards to the accuracy and the cross-entropy loss.

Model	Accuracy_Ttraining Set	Accuracy_Valid Set	Cross Entropy_Training Set	Cross Entropy_Valid Set
Mobilenet_0.25_224	86.23%	85.16%	0.6317	0.7883
Mobilenet_0.5_224	91.32%	90.24%	0.3023	0.3783
Mobilenet_0.75_224	92.49%	92.73%	0.2149	0.2439
Mobilenet_1.0_224	94%	93%	0.1307	0.2450
Inception-V3	96.05%	95.66%	0.1164	0.1346

**Table 11 diagnostics-10-00131-t011:** Comparative results of the proposed models and other models used with different dataset.

Model	Accuracy (%)	Senstivity (%)	Specificity (%)	Dataset
Linear classifier	66.5	65.2	67.2	Kaggle
Vanilla 3DCNN	70.5	59.3	76.1	Kaggle
3D AlexNet	85.79	82.74	88.04	Kaggle
3D-Googlenet	87.95	82.74	91.61	Kaggle
DFCNet [24]	86.02	80.91	83.22	LIDC-IDRI
TumorNet [42]	87.41	81.70	85.17	LIDC-IDRI
CMixNet [17]	88.79	93.97	89.83	LIDC-IDRI
straight 3D-CNN + softmax classifier [16]	90.23	86.40	93.09	Kaggle
Hybrid 3D-CNN + RBF-based SVM [16]	91.8	88.53	94.23	Kaggle
3D CMixNet + GBM [17]	91.13	–	–	LIDC-IDRI
3D CMixNet + GBM + Biomarkers [17]	94.17	94	91	LIDC-IDRI
Ours: DeeplabV3plus(ex_65) + Mobilenet-V1_1.0_224	93	90.01	94.3	Kaggle
Ours: DeeplabV3plus(ex_65) + Inception-V3	95.66	91.2	97.24	Kaggle

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
