# Peer review of "A Two-Stage Framework for Automated Malignant Pulmonary Nodule Detection in CT Scans"

_diagnostics, 2020, doi:10.3390/diagnostics10030131_

Round 1

Reviewer 1 Report

The authors demonstrate that  the DeepLab model for semantic segmentation, significantly improves the accuracy of nodule detection compared to the classical U-Net model and its most recent variants. However the authors did not address the percent of false diagnosis with their model. This has to be included in their study.

Author Response

A Two-Stage Framework for Automated
Malignant Pulmonary Nodule Detection in CT Scans
Response to Reviewer 1 Comments
Editor – We would first like to thank you and the reviewers for your time and efforts in providing constructive feedback on our article. Please see below our responses to the reviewer’s comments.
In the modified version of the article, all the modifications are shown in red.
Point 1: The authors did not address the percent of false diagnosis with their model. This has to be included in their study.
Response 1: Thank you for pointing this out. We have added a column in Table 6 showing the FP/Scan for various recently proposed algorithms including ours. It is worth mentioning that we have added several references to the table, compared to the table in the first version of the article. Including the FP/Scan highlights another merit for the proposed framework. Finally, we have added another table (Table 7 in the modified version of the article) to indicate the confusion matrix of the adopted model.

Reviewer 2 Report

This paper introduced a new pipeline fro malignant pulmonary nodule detection (PND) in CT scans. The authors combined a sequence of modern Convolution network algorithms to improve the results. They compared the proposed algorithm to multiple modified U-Net. 

The biggest concern is the comparisons to general U-net may not be fair on PND CT detection. There are many machine learning algorithms that have been successfully applied to PND (e.g., Automated Lung Nodule Detection and Classification Using Deep Learning Combined with Multiple Strategies, 2019; Classification of Lung Nodules in CT Scans Using Three-dimensional Deep Convolutional Neural Networks with a Checkpoint Ensemble Method, 2018). I would suggest the authors should compare the proposed algorithms to them. 

Author Response

A Two-Stage Framework for Automated
Malignant Pulmonary Nodule Detection in CT Scans
Response to Reviewer 2 Comments
Editor – We would first like to thank you and the reviewers for your time and efforts in providing constructive feedback on our article. Please see below our responses to the reviewers’ comments.
In the modified version of the article, all the modifications are shown in red.
Point 1: The biggest concern is the comparisons to general U-net may not be fair on PND CT detection. There are many machine learning algorithms that have been successfully applied to PND (e.g., Automated Lung Nodule Detection and Classification Using Deep Learning Combined with Multiple Strategies, 2019; Classification of Lung Nodules in CT Scans Using Three-dimensional Deep Convolutional Neural Networks with a Checkpoint Ensemble Method, 2018). I would suggest the authors should compare the proposed algorithms to them.
Response 1: Thank you for pointing this out. We have added more comparisons to other techniques in the literature including the references mentioned by the respected reviewer [17, 18]. This is indicated in Table 6 in which we have also extended the comparisons to include the False Positives/Scan (FP/Scan), which further highlight the merits of the proposed framework. Moreover, we have added another table (Table 7 in the modified version of the article) to indicate the confusion matrix of the adopted model. Furthermore, in Table 11, we have included reference [17], and we show that: While it attains higher sensitivity, our adopted model attains higher accuracy and specificity.
It is worth mentioning that in [17], the authors included other features, such as physiological symptoms and clinical biomarkers, in order to enhance the accuracy. Our proposed framework achieved a higher accuracy without incorporating physiological symptoms and clinical biomarkers. This makes the proposed framework more data-efficient.

Round 2

Reviewer 1 Report

The authors  evaluate the performance of the semantic segmentation stage by adopting two network backbones, namely, MobileNet-V2 and Xception for PND. They show a mIoU increase of 60% and a dice coefficient increase of 30% compared to U-Net. On LUNA16, the two-stage framework attained a sensitivity of 96.4%, outperforming other recent models in the literature, including deep models. adopting a transfer learning approach outweights of the
first stage of the framework, to infer binary (malignant-benign) labels on the Kaggle dataset. However, the authors did not describe how they remove outliers, the limitations of their model and how it works with control patients. Obviously the images in the database are already selected for improvement of the analysis.

Reviewer 2 Report

The authors added new experiences to compare with two PND CT detection methods. However, they only list sensitivity and FP in Table 7. Since AUC (area under the curve) or Accuracy are better overall evaluations, I would like to see their comparison results in Table 7.

Round 3

Reviewer 2 Report

I am satisfied with this revision.